



# Parallel Implementation of the SHYFEM Model

Giorgio Micaletto[1], Ivano Barletta[1,3], Silvia Mocavero[1], Ivan Federico[1], Italo Epicoco[1,2], Giorgia Verri[1], Giovanni Coppini[1], Pasquale Schiano[1], Giovanni Aloisio[1,2], and Nadia Pinardi[3]

[1]Euro-Mediterranean Centre on Climate Change Foundation, via Augusto Imperatore 16, 73100 Lecce, Italy
[2]Department of Engineering for Innovation, University of Salento, via per Monteroni, 73100 Lecce, Italy
[3]Department of Physics and Astronomy, University of Bologna, Italy

**Correspondence:** Italo Epicoco (italo.epicoco@unisalento.it)

**Abstract.**

This paper presents the MPI-based parallelization of the three-dimensional hydrodynamic model SHYFEM (System of HydrodYnamic Finite Element Modules). The original sequential version of the code was parallelized in order to reduce the execution time of high-resolution configurations using state-of-the-art HPC systems. A distributed memory approach was used, based on the message passing interface (MPI). Optimized numerical libraries were used to partition the unstructured grid (with a focus on load balancing) and to solve the sparse linear system of equations in parallel in the case of semi-to-fully implicit time stepping. The parallel implementation of the model was validated by comparing the outputs with those obtained from the sequential version. The performance assessment demonstrates a good level of scalability with a realistic configuration used as benchmark.

## 1 Introduction

Ocean sciences are significantly supported by numerical modeling, which help to understand physical phenomena or provide predictions both in the short term or from a climate perspective. The reliability of ocean prediction is strictly linked to the ability of numerical models to capture the relevant physical processes.

The physical processes taking place in the oceans occupy a wide range of spatial and time scales. The ocean circulation is highly complex, in which physical processes at large scales are transferred to smaller scales, resulting in mesoscale and sub-mesoscale structures, or eddies (Hallberg, 2013).

The coastal scale is also rich in features driven by the interaction between the regional scale dynamics and the complex morphology typical of shelf areas, tidal flats, estaurines and straits.

In both large-scale and coastal modeling, the spatial resolution is a key factor.

Large-scale applications require a finer horizontal resolution than the Rossby radius (in the order of 100km in mid-latitudes and less than 10km in high latitudes) to capture the mesoscale (Maltrud and McClean, 2005). The geometry of the domain drives the selection of spatial resolution in the coastal environment, where grid spacing can be in the order of meters.

Ocean circulation models use mostly structured meshes, and have a long history of development (Griffies et al., 2010) often organized in a community model framework.



In the last few decades, however, the finite volume (Casulli and Walters, 2000)(Chen et al., 2003) or the finite element approach (Danilov et al., 2004) (Zhang et al., 2016) (Umgiesser et al., 2004), applied to unstructured meshes, has become more common, especially in the coastal framework, where the flexibility of the mesh particularly suits the complexity of the environment. Applications of unstructured grids include the modeling of storm surges (Westerink et al., 2008), ecosystem modeling (Zhang et al., 2020b), sediment transport (Stanev et al., 2019), and flow exchange through straits (Stanev et al.,

30   2018).

Representing several spatial scales in the same application renders the unstructured grid appealing in simulations aimed at bridging the gap between the large scale flow and the coastal dynamics (Ringler et al., 2013). Advances in the numerical formulation have meant that unstructured models can also be used for large-scale simulations that address geostrophic adjustments and conservation properties comparably to regular grids (Griffies et al., 2010), thus leading to simulations that are suitable for

climate studies (Danilov et al., 2017) (Petersen et al., 2019).

The computational cost of a numerical simulation depends on the order of accuracy of the numerical scheme and the grid scale (Sanderson, 1998). In the case of structured grids, the computational cost increases inversely on the grid space with the power of the dimensions represented by the model. Estimating the computational cost in unstructured grids is not as straightforward as in regular grids, but it is commensurate to the latter when low order schemes are used (Ringler et al., 2013)

(Koldunov et al., 2019).

Both large-scale and coastal applications may involve significant computational resources because of the high density of mesh descriptor elements required to resolve dominant physical processes. The computational cost is also determined by upper limits on the time step, making meaningful simulations prohibitive for conventional machines. Access to HPC resources is essential for performing state-of-the-art simulations.

Several successful modelling studies have involved the SHYFEM unstructured grid model (Umgiesser et al., 2004) (Bellafiore and Umgiesser, 2010) in the development of regional (Ferrarin et al., 2019) (Bajo et al., 2014) (Ferrarin et al., 2018) and coastal applications (Umgiesser et al., 2014).

The range of applications of SHYFEM was recently extended in a multi-model study to assess the hazards related to climate scenarios (Torresan et al., 2019), the change in sea level in tidal marshes in response to hurricanes (Park et al., 2021) and in a

high-resolution simulation of the Turkish Strait System dynamics under realistic atmospheric and lateral forcing (Ilicak et al., 2021).

SHYFEM was also applied to produce seamless three-dimensional hydrodynamic short-term forecasts on a daily basis (Federico et al., 2017) (Ferrarin et al., 2019) from large to coastal scales. SHYFEM has also been used in relocatable mode (Trotta et al., 2021) to support emergency responses to extreme events and natural hazards in the world's oceans. Both in

forecasting systems and relocatable services, the need for reduced computational costs is crucial in order to deliver updated forecasts with a longer time window.

We implemented a version of the SHFYEM code that can be executed on parallel architectures, addressing the problem of load balancing that is strictly related to the grid partitioning, the parallel scalability and inter-node computational overhead. Our aim was to make all these applications (study process simulation at different scales, long-term and climatic implementa-





tions, forecasting and relocatable systems) practical, also from a future perspective where the computational cost is constantly increasing with the complexity of the simulations. We adopted a distributed memory approach, with two key advantages: (i) reduction in runtime with the upper limit determined by the user's choice of resources (ii) memory scalability, allowing for highly memory-demanding simulations.

The distributed memory approach, based on the Message Passing Interface (The MPI Forum, 1993), can coexist with the

shared memory approach and is widely used to parallelize unstructured ocean models, such as MPAS (Ringler et al., 2013) and FESOM2 (Danilov et al., 2017), which are devised for global configurations. MPI codes that address coastal processes include SLIM3D (Kärnä et al., 2013), SCHISM (Zhang et al., 2016), and FVCOM (Chen et al., 2003) (Cowles, 2008).

The MPI developments carried out in this work consist of additional routines that wrap the native MPI directives, without undermining the code readability. Some aspects of the parallel development, such as the domain decomposition and the solution

of free surface equations, were achieved using external libraries.

Section 2 introduces the SHYFEM model and its main features. Section 3 describes the methodology and the design of the distributed memory parallelization, through the partitioning strategy and management of the data dependencies among the MPI processes. Section 4 describes the implementation; Section 5 presents the model validation and the performance assessment. Finally, the conclusions are drawn in the last section.

## 75  2   The SHYFEM model

SHYFEM solves the ocean primitive equations, assuming incompressibility in the continuity equation, and advection-diffusion equation for active tracers using finite element discretization based on triangular elements (Umgiesser et al., 2004). The model uses a semi-implicit algorithm for the time integration of the free surface equation, which makes the solution stable by damping the fastest external gravity waves. The Coriolis term and pressure gradient in the momentum equation, and the divergence terms

in the continuity equation are treated semi-implicitly. The vertical eddy viscosity and vertical eddy diffusivity in the tracer equations are treated fully implicitly for stability reasons. The advective and horizontal diffusive terms in the momentum and tracer equations are treated explicitly. Smagorinsky's formulation (Blumberg and Mellor, 1987) (Smagorinsky, 1963) is used to parameterise the horizontal eddy viscosity and diffusivity. To compute the vertical viscosities, a turbulence closure scheme was used. This scheme is adapted from the k-epsilon module of GOTM (General Ocean Turbulence Model) described in (Burchard

and Petersen, 1999). A detailed description of the 3D model equations is given in (Bellafiore and Umgiesser, 2010) and (Maicu et al., 2021).

### 2.1   Time Discretization

SHYFEM equations are discretized in time with a forward time stepping scheme with terms that are evaluated at time level n+1 with weight $0 \leq \theta \leq 1$ (and time level n terms with weight 1-$\theta$) the equations of hydrodynamics. The time level n+1 of

external pressure gradient in the momentum equations has weight $\gamma \in [0, 1]$ and the time level of the divergence of barotropic flow in the continuity equation has weight $\beta \in [0, 1]$. When one of $\gamma$ or $\beta$ is 0, the scheme is considered explicit in time.





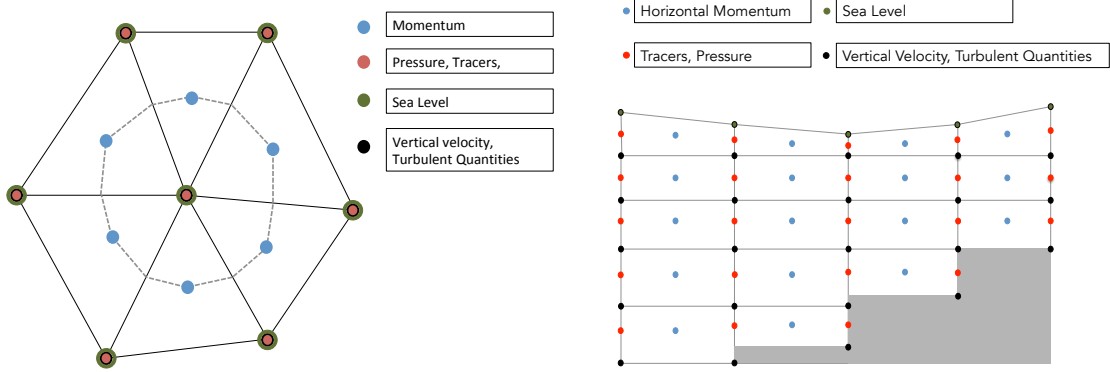

**Figure 1.** Example of Variables on the SHYFEM Horizontal/Vertical Grid **[Left]** Sketch of horizontal connectivity of variables. **[Right]** Distribution of variables on the vertical grid.

The treatment of external pressure gradient and divergence in continuity is consistent with the method described in (Campin et al., 2004) and used in the pressure method of MITgcm (Marshall et al., 1997).

SHYFEM applies a semi-implicit scheme to the Coriolis force and vertical viscosity with weights of time level n+1 $a_F$ and

$a_T$, respectively. As described in (Campin et al., 2004), the solution requires a time step for the momentum from time level n to the intermediate level *. Using the notation reported in table (1), the velocity equation integrated over a generic layer $l$ is

$$\left(\frac{\boldsymbol{U}_l^* - \boldsymbol{U}_l^n}{\Delta t}\right) + a_F \Delta t f \hat{\boldsymbol{k}} \times \left(\frac{\boldsymbol{U}_l^* - \boldsymbol{U}_l^n}{\Delta t}\right) - a_T \Delta t D_z \left(\frac{\boldsymbol{U}_l^* - \boldsymbol{U}_l^n}{\Delta t}\right) = \boldsymbol{F}_l^n - f \hat{\boldsymbol{k}} \times \boldsymbol{U}_l^n + D_z \boldsymbol{U}_l^n - g h_l^n \boldsymbol{\nabla} \eta^n \tag{1}$$

where $D_z$ is the vertical viscosity operator and the term $\boldsymbol{F}_l^n$ contains advection, horizontal turbulent viscosity and pressure terms

$$\boldsymbol{F}_l^n = (\boldsymbol{u}_l^n \nabla) \boldsymbol{U}_l^n + \int_{z_l}^{z_{l-1}} w^n \frac{\partial \boldsymbol{u}^n}{\partial z} \mathrm{d}z + A_H^l \nabla^2 \boldsymbol{U}_l^n + \frac{h_l}{\rho_0} \nabla \mathrm{p}_{\mathrm{atm}} + \frac{g h_l}{\rho_0} \int_{z_{l-1/2}}^{\eta^n} \nabla \rho' \mathrm{d}z \tag{2}$$

The implementation of $D_z$ and the vertical advection are given in Appendix B. The vertical viscosity in the momentum equation is commonly treated implicitly in numerical ocean models. This is because the inversion of the tri-diagonal matrix associated with the system is not computationally demanding.

SHYFEM considers the semi-implicit treatment of the Coriolis term, since it can lead to instability when the friction is too

low. The weights assigned to time level n and n+1 of this term in the ocean models that perform semi-implicit treatment are commonly 0.5/0.5, which is the most accurate scheme in the representation of inertial waves (Wang and Ikeda, 1995).





The vertical viscosity terms in equations (1) introduce a dependency between adjacent layers, and the equations cannot be inverted trivially. The Coriolis term, also introduces dependency between zonal and meridional momentum, requiring a simultaneous inversion of the two equations.

The subsequent linear system is sparse with a penta-diagonal structure with dimension $2\,l_{max} \times 2\,l_{max}$, where $l_{max}$ is the number of active layers of the element. The system is solved in each element by Gauss elimination with partial pivoting.

The solution of equation (1) gives the tendency of momentum $\Delta\boldsymbol{U} = (\boldsymbol{U}^* - \boldsymbol{U}^n)/\Delta t$ which is used to calculate the momentum at time level *

$$\boldsymbol{U}^* = \boldsymbol{U}^n + \Delta t\Delta\boldsymbol{U} \tag{3}$$

The elliptic equation of the prediction of free surface $\eta$ is

$$\eta^{n+1} + \delta\nabla\big(H\nabla\eta^{n+1}\big) = \eta^n + \delta\nabla\big(H\nabla\eta^n\big) + \Delta t\nabla\big(\beta\bar{\boldsymbol{U}}^* + (1-\beta)\bar{\boldsymbol{U}}^n\big) + \Delta t(P-E) \tag{4}$$

with $\delta = g\gamma\beta\Delta t^2$, $P - E$ the freshwater flux at the surface and $H = \sum_l h_l^n$, $\bar{\boldsymbol{U}}^* = \sum_l \boldsymbol{U}_l^*$, $\bar{\boldsymbol{U}}^n = \sum_l \boldsymbol{U}_l^n$ respectively. The solution of equation (4) is described in section (4.3).

The advancement of momentum equations is finalized with the correction step, using $\eta^{n+1}$

$$\boldsymbol{U}_l^{n+1} = \boldsymbol{U}_l^* - g\Delta t\gamma h_l^n\nabla\left(\eta^{n+1} - \eta^n\right) \tag{5}$$

Vertical velocities at time level n+1 $w^{n+1}$ are diagnosed with the continuity equation for the control volume associated with a node $k$ (fig. 1a)

$$\delta_{l1}\frac{\partial h_l}{\partial t} + \left[\boldsymbol{\nabla}\tilde{\boldsymbol{U}}_l + Q_l/A\right] + \left(w_{l-1}^{n+1} - w_l^{n+1}\right) = 0 \tag{6}$$

with the vertical discretization shown in fig. (C1) and $\tilde{\boldsymbol{U}}_l = \beta\boldsymbol{U}_l^{n+1} + (1-\beta)\boldsymbol{U}_l^n$ and $\delta_{l1}$ indicating that the thickness is variable only for surface layer. The quantity $Q_l$ [m³/s] represents mass fluxes from surface or from internal sources, thus the equation contains the area $A$ of the control volume. The equation is integrated from the bottom with the boundary condition $w_{l_{max}}^{n+1} = 0$.

Advection-diffusion equation for a generic tracer $T$ is solved treating the vertical diffusion implicitly (default value for $a_D$ = 1). The vertical advection can be treated semi or fully implicitly in the case of upwind scheme (default value $a_V = 0$)





$$\frac{h_l^{n+1}T_l^{n+1} - h_l^n T_l^n}{\Delta t} = -\nabla(\tilde{\boldsymbol{U}}T^n)_l - a_V(w_t T_t - w_b T_b)_l^{n+1} - (1-a_V)(w_t T_t - w_b T_b)_l^n \tag{7}$$

$$+ h_l^n K_H \nabla^2 T_l^n + a_D\left[K_V^n \frac{\partial T^{n+1}}{\partial z}\bigg|_{l-1} - K_V^n \frac{\partial T^{n+1}}{\partial z}\bigg|_l\right] \tag{8}$$

$$+ (1-a_D)\left[K_V^n \frac{\partial T^n}{\partial z}\bigg|_{l-1} - K_V^n \frac{\partial T^n}{\partial z}\bigg|_l\right] + Q \tag{9}$$

where subscripts $_t$ and $_b$ indicate the value of the tracer at the top and bottom of the layer, respectively.

The horizontal diffusivity follows Smagorinsky's formulation. $K_V$ represents the background molecular diffusivity plus the
turbulent diffusivity (always at time step n) and $Q$ source / sink term, described in (Maicu et al., 2021). The equation (7) is
solved with the Thomas algorithm for tridiagonal matrices in each node column.

The density is updated by means of the equation of state (EOS) under the hydrostatic assumption

$$\rho_l = \text{EOS}(T_l^{n+1}, S_l^{n+1}, p_l) \tag{10}$$

$$p_l = g\left(\sum_{k<l}(\rho_0 + \rho_k')h_k + 1/2(\rho_0 + \rho_k')h_l\right) \tag{11}$$

where the EOS can either be the UNESCO EOS80 (Fofonoff and Millard, 1983) or the EOS from (Jackett and McDougall,
1997).

The sub-steps of the SHYFEM solution method are in Algorithm (1) with the corresponding equations.

---

**Algorithm 1** SHYFEM Time loop

---

**while** $t <$ num_timesteps **do**

    AdvanceMomentum                {eq. (1)}

    SolveBarotropicEquation    {eq. (4)}

    FinaliseMomentum             {eq. (5)}

    CalcVerticalVelocity        {eq. (6)}

    SolveTracerAdvection      {eq. (7)}

    UpdateDensity               {eq. (10)}

**end while**

---

## 2.2 Spatial Discretization and Dependencies

Figure (1a) shows how the model variables are staggered over the computational grid. Horizontal momentum (U,V) are located
in the element centers, while all the others are located on the vertexes (vertical velocity $w$ and scalars). Each vertex has a
corresponding finite volume (dashed lines in fig 1a). The staggering of hydrodynamic variables is essential to have a mass-
conserving model (Jofre et al., 2014) (Felten and Lund, 2006).





Variables are also staggered in the vertical grid, as shown in figure (1b).

The turbulent and molecular stresses and the vertical velocity are computed at the bottom interface of each layer (black dots
in fig. 1b), the free surface is at the top of the upper layer thus determining the variable volume of the top cells, all the other
variables are defined at the layer center (red dots in fig. 1a).

Scalar variables (red) are staggered with respect to vertical velocity (black), referenced in the middle and at layer interfaces,
respectively. The sea surface elevation is a 2D field defined only in the w points at surface.

The grid cells on the top layer can change their volume as a result of the oscillation of the free surface. The number of active
cells along the vertical direction depends on the sea depth.

The spatial discretization of the governing equations in the FEM framework is based on the assumption that the approximate
solution is a linear combination of shape functions defined in the 2D space. In this section we provide suggestions for the
practical use of the FEM method in the relevant terms of the SHYFEM equations.

Table (2) reports the possible connectivities that arise from the physics of the SHYFEM equations. The table also shows
the definition for the partial derivatives of the linear shapefunctions to calculate horizontal gradients of scalar quantities and
divergence of vector fields. Appendix A gives more insights into the shapefunctions and the calculation of their gradients.

Considering a scalar field, such as the surface elevation $\eta$ in eq. (1), its gradient is a linear combination of the gradient of
shapefunctions.

$$\left(\frac{\partial \eta}{\partial x}, \frac{\partial \eta}{\partial y}\right)_{e} = \left(\sum_{k \in [e]} \partial_x \phi(k,e)\eta(k), \sum_{k \in [e]} \partial_y \phi(k,e)\eta(k)\right) \tag{12}$$

The resulting gradient is referenced to the element e, and is a combination of values referenced to its nodes. This means that
the calculation of the horizontal gradient of a scalar field consists of a *node-to-element* dependency (fig. 3C).

Considering a vector field, such as $\boldsymbol{U}$, its horizontal divergence is a linear combination of the same gradients as the shape-
functions, as detailed below:

$$(\nabla \boldsymbol{U})_k = \sum_{e \in [k]} \partial_x \phi(k,e)U(e) + \partial_y \phi(k,e)V(e) \tag{13}$$

where the divergence of the horizontal velocity field is referenced to the node k and is the sum of the momentum fluxes
inside/outside the node control volume from the elements surrounding the node $e \in [k]$. Calculation of the divergence of vector
fields consists of a *element-to-node* dependency (fig. 3b).

The horizontal viscosity stress in x direction (and similarly in y direction), has the form

$$A_H(e) \sum_{e' \in [e]} \frac{U(e') - U(e)}{(A(e') + A(e))/2} \tag{14}$$

where the contributions come from the differences between the momentum of the current element $U(e)$ and the momentum
of surrounding elements $e'$ that share one side with $e$ divided by the sum of areas of both elements $A(e) + A(e')$. $A_H(e)$



| name | description | unit | index range |
|---|---|---|---|
| nel,nkn,nlv | total number of elements/nodes/layers | NA | NA |
| $l$ | vertical layer index | NA | 1:nlv |
| e | element index | NA | 1:nel |
| k | node index | NA | 1:nkn |
| $z_l$ | depth of interface $l$ | m | 0:nlv |
| $h_l(e) = z_{l-1} - z_l$ | thickness of element e at layer $l$ | m | l=1:nlv,e=1:nel |
| $\boldsymbol{u}_l(e) = (u_l(e),v_l(e))$ | velocity vector in element e at layer $l$ | m/s | l=1:nlv,e=1:nel |
| $\boldsymbol{U}_l(e) = (U_l(e),V_l(e)) =$ $= (u_l(e)h_l(e),v_l(e)h_l(e))$ | layer integrated velocity vector in element e at layer $l$ | m²/s | l=1:nlv,e=1:nel |
| $w_l(k)$ | vertical velocity at node k at interface $l$ | m/s | l=0:nlv,k=1:nkn |
| $T_l(k),S_l(k)$ | Temperature/Salinity at node k at layer $l$ | C°/PSU | l=1:nlv,k=1:nkn |
| $\nu_l(k),K_V^l(k)$ | vertical turbulent viscosity/diffusivity at node k at interface $l$ | m²/s | l=0:nlv,k=1:nkn |
| $\eta(k)$ | free surface elevation at node k | m | k=1:nkn |
| $\rho_0$ | reference density | Kg/m³ | NA |
| $\rho_l'(k)$ | deviation of density from reference | Kg/m³ | l=1:nlv,k=1:nkn |
| $\nabla$ | 2D nabla operator | 1/m | NA |
| $\hat{\boldsymbol{k}}$ | vertical unit vector | NA | NA |
| $\Delta t$ | model time step | s | NA |
| $f(e)$ | Coriolis parameter in element e | 1/s | e=1:nel |
| $A_H^l(e)$ | Horizontal viscosity coefficient at element e at layer $l$ | m²/s | l=1:nlv,e=1:nel |
| $\gamma,a_T,a_F$ | weight of time level n+1 for external pressure gradient, vertical viscosity and Coriolis in momentum | NA | NA |
| $\beta$ | weight of time level n+1 for horizontal divergence in continuity equation | NA | NA |
| $a_V,a_D$ | weight of time level n+1 for vertical advection and diffusion in tracers equation | NA | NA |

**Table 1.** Notations adopted in this work

is the viscosity coefficient calculated according to (Smagorinsky, 1963). The viscosity stress components thus consist in an *element-to-element* dependency (fig. 3a).

The equation (4) can be seen in matrix for $\mathbf{A}\eta^{n+1} = \mathbf{B}$ with $\mathbf{A} = I + \delta \nabla \cdot (H\nabla)$ and $\mathbf{B}$ containing the right-hand side of

(4). The left hand side of (4) is discretized as follows:

$$\eta^{n+1}(k) + \delta \sum_{e \in [k]} \left( \partial_x \phi(k,e) H_e \sum_{k' \in [e]} \partial_x \phi(k',e) \eta^{n+1}(k') + \partial_y \phi(k,e) H_e \sum_{k' \in [e]} \partial_y \phi(k',e) \eta^{n+1}(k') \right) \quad (15)$$





| k $\in$ [e] | nodes k of element e (fig. 3c) |
|---|---|
| e $\in$ [k] | elements e around node k (fig. 3b) |
| e' $\in$ [e] | elements e' sharing an edge (fig. 3a) with element e |
| k' $\in$ [k] | nodes k' around node k (fig. 3d) |
| $\partial_x\phi$(k,e),$\partial_y\phi$(k,e) | gradients of $\phi$ shapefunction for node k in element e |

**Table 2.** Notation for connectivity and gradient of shapefunctions.

creating a dependency between node $k$ and the surrounding nodes $k' \in [k]$.

The terms $1 + \delta(\partial_x\phi(\text{k,e})H_e\partial_x\phi(\text{k',e})$ and $\partial_y\phi(\text{k,e})H_e\partial_y\phi(\text{k',e}))$ represent diagonal entries if $k' == k$ and terms $\delta(\partial_x\phi(\text{k,e})H_e\partial_x\phi(\text{k',e})$ and $\partial_y\phi(\text{k,e})H_e\partial_y\phi(\text{k',e}))$ represent off-diagonal entries in case $k' \neq k$.

The dependency between adjacent nodes of $\eta$ introduced by the discretization of **A** is of the kind *node-to-node* (fig. 3d).

## 3 Parallel approach

Scientific and engineering numerical simulations involve an ever-growing demand for computing resources due to the increasing model resolution and complexity. Computer architectures satisfy simulation requirements through a variety of computing hardware, often combined together into heterogeneous architectures. There are key benefits from the design (or re-design) of a parallel application (Zhang et al., 2020a; Fuhrer et al., 2014) and the choice of parallel paradigm, taking into account the features of computing facilities (Lawrence et al., 2018). Shared and distributed parallel programming can be mixed to better exploit heterogeneous architectures. MPI+X enables the code to be executed on clusters of NUMA nodes equipped with CPUs, GPUs, accelerators, etc.. This mixing should be done by taking into account the main features of the two paradigms. The shared memory approach enables multiple processing units to share data but does not allow the problem to be scaled on more than one computing node, setting an upper bound to the available memory. The distributed memory approach, on the other hand, enables each computing process to access its own memory space, so that bigger problems can be addressed by scaling the memory over multiple nodes. However, communication between parallel processes is needed in order to satisfy data dependencies. Given that configurations will require even more memory and computing power, the strategy used to parallelize the SHYFEM model is based on the distributed memory approach with MPI. The parallelization strategy can be easily combined with the existing shared memory implementation based on OpenMP (Pascolo et al., 2016), or with other approaches not yet implemented (e.g. OpenACC).

### 3.1 The domain partitioning issue

Identifying data dependencies is key for the design of the parallel algorithm, since inter-process communications need to be introduced to satisfy these dependencies.





In the case of a structured grid, each grid point usually holds information related to the cell discretized in the space, and data
dependencies are represented by a stencil containing the relations between each cell and its neighbours. For example, we could
have five or nine point stencils that represent the dependencies of the current cell with regards to its four neighbours north,
south, east and west, or alternatively cells along diagonals could also be considered for computation.

On the other hand, unstructured grid models can be characterized by dependencies among nodes (the vertexes), elements
(triangles) or nodes and elements. These kinds of dependencies need to be taken into account when the partitioning strategy is
defined.

## 3.2    Partitioning strategy

The choice between element-based or node-based partitioning, see Figure 2, aims to reduce the data exchange among different
processes.

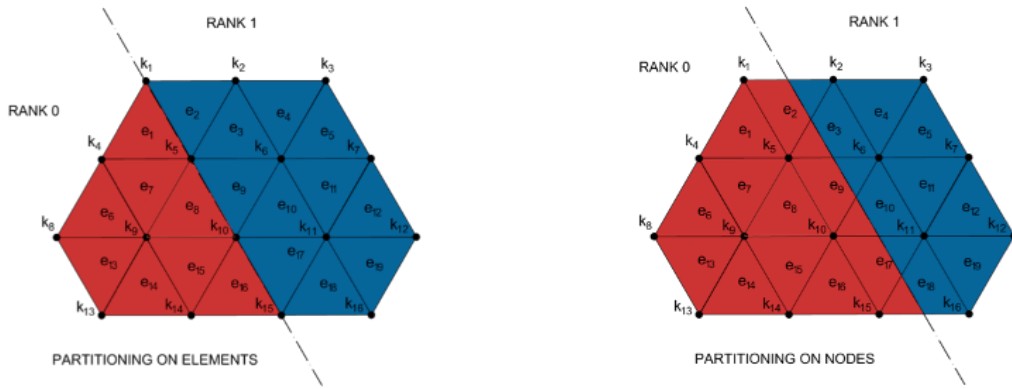

**Figure 2.** Element based partitioning and Node based partitioning.

The best partitioning strategy cannot be defined absolutely. In fact, it usually depends on the code architecture and its
implementation.

Analysis of the SHYFEM code shows that an element-based domain decomposition minimizes the number of communica-
tions among the parallel processes. Four types of data dependencies (see Figure 3) were identified within the code: *element-to-
element* (A): the computation on each element depends on the three adjacent elements; *element-to-node* (B): the node receives
data from the incident elements (usually six); *node-to-element* (C): the element needs data from its three nodes; *node-to-node*
(D): the computation on each node depends on the adjacent nodes.

The element-based partitioning needs data exchange when there are dependencies A, B and D, while node-based partitioning
needs data exchange with dependencies C and D. The data dependency A happens only when momentum is exchanged to
compute the viscosity operator; the data dependency D happens in two cases when matrix-vector products are computed for



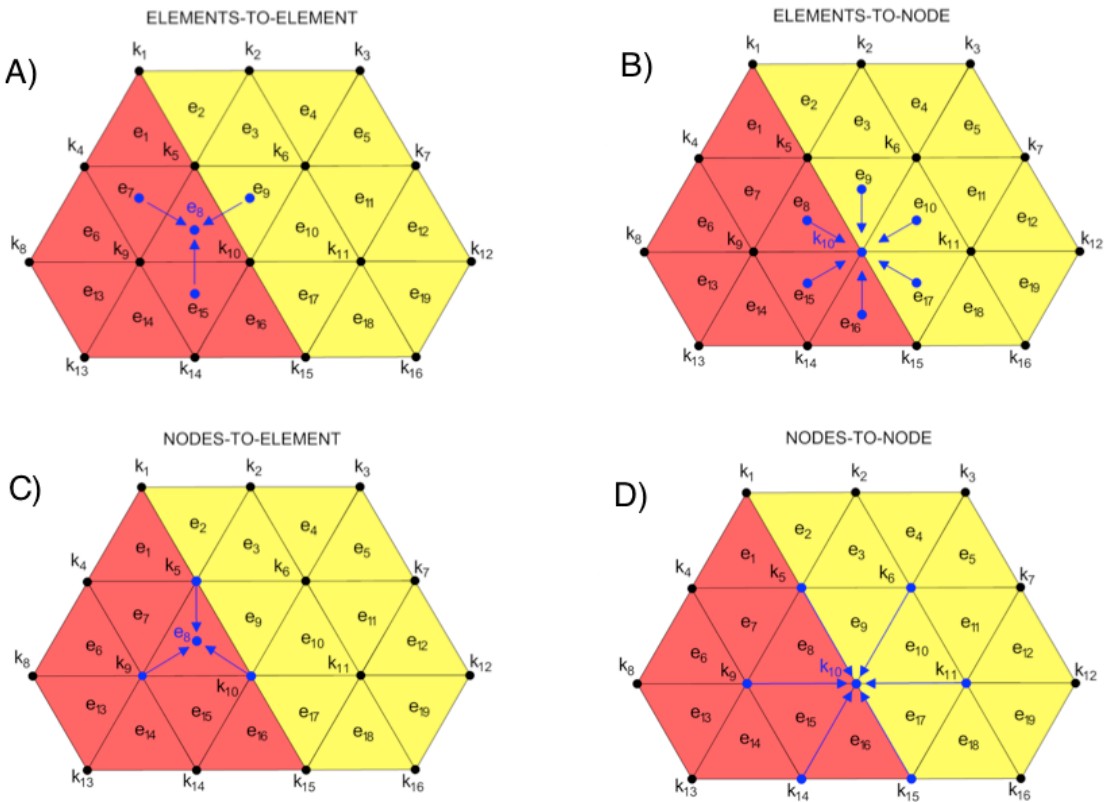

**Figure 3.** Possible data dependencies on a staggered grid

the implicit solution of the Free Surface Equation (FSE) which is solved using the PETSc external libraries; finally, the data
dependency C is the most common in the code, more frequent than dependency B.

We can thus summarize that, after analysing the SHYFEM code, element-based partitioning reduces the data dependencies
that need to be solved through data exchanges between neighbouring processes. Clearly, the computation on nodes shared
among different processes is replicated.

## 3.3 The partitioning algorithm

The second step, after selecting the partitioning strategy, is to define the partitioning algorithm. This represents a way of dis-
tributing the workload among the processes for an efficient parallel computation. The standard approach (Hendrickson and
Kolda, 2000) is to consider the computational grid as a graph, and to apply a graph partitioning strategy in order to distribute
the workload. There are several parallel tools that provide solutions to the partitioning problem, such as ParMetis (G. Karypis,
1997), the parallel extended version of Metis (G. Karypis, 1999), Jostle (Walshaw and Cross, 2007), PT-Scotch (Chevalier and
Pellegrini, 2008), and Zoltan (Hendrickson and Kolda, 2000). The scalability of the Zoltan PHG partitioner (Sivasankaran Ra-





jamanickam, 2012) was a key factor in the choice of the partitioning tool for the parallel version of SHYFEM. The Zoltan library simplifies the development and improves the performance of the parallel applications based on geometrically complex grids. The Zoltan framework includes parallel partitioning algorithms, data migration, parallel graph colouring, distributed data directories, unstructured communication services, and memory management packages. It is available as open source software. An offline static partition module was designed and implemented. It is executed once before beginning the simulation, when the number of parallel processes has been decided. The partitioning phase aims to minimize the inter-process edge cuts and the differences among the workloads assigned to the various processes. The weights used in the graph partitioning for each element is proportional to the number of vertical levels of the element itself.

## 4 Parallel code implementation

The SHYFEM code has a modular structure. It enables users to customize the execution by changing the parameters defined within a configuration file (i.e. namelist), to set up the simulation, and to activate the modules that solve hydrodynamics, thermodynamics, turbulence.

This section details the changes made to the original code, introducing the additional data structures needed to handle the domain partitioning, the MPI point-to-point and collective communications, the solution of the FSE using the external PETSc library, and the I/O management.

### 4.1 Local-Global mapping

The domain decomposition over several MPI processes entails mapping the information of local entities to global entities. The entities to be mapped are the elements and nodes. As a consequence of the partitioning procedure, each process holds two mapping tables, one for the elements and one for the nodes. The mapping table of the elements stores the correspondence between the global identifier of the element (which is globally unique) with a local identifier of the same element (which is locally unique). The same also happens with the mapping table associated with the nodes. Mapping information is stored within two local data structures, containing the Global IDentification Number (GID) of elements and nodes respectively. The order of GID elements in local structures is natural, namely they are set in ascending order of GIDs. The local-global mapping is represented by the position in the local structure of the GIDs, called Local IDentification Number (LID).

The GIDs of the nodes are stored in a different order. The GIDs of the nodes that belong to the boundary of the local domain are stored at the end of the mapping table. This provides some computational benefits: it is easy to identify all of the nodes on the border; in most cases it is better to first execute the computation over all of the nodes in the inner domain and after the computation, over the nodes at the boundary; during the data exchange it is easy to identify which nodes need to be sent and which ones need to be updated with the data from the neighbouring processes.





## 4.2 Data exchange

Data exchanges are executed when element-to-node and element-to-element dependencies happen and MPI point-to-point communications are used. In the first case, each process receives information based on the elements that share the target node from the processes the elements belong to. It keeps track of the shared nodes in terms of numbers and LIDs. Each process

computes the local contribution and sends it to the interested neighbouring processes. The information received is stored in a temporary 3D data structure defined for nodes, vertical levels and processes. A reduction operation is performed on the information received.

The element-to-element dependency happens only once in the time loop required to compute the viscosity operator. In this case, each process sends its contribution to the neighbours in terms of momentum values. Each process keeps track of

the elements to be sent/received in terms of the numbers and LIDs, using two different data structures. In this case, the data structure extends the local domain in order to include an overlap used to store the data received from the neighbours as shown in figure (4). Non-blocking point-to-point communications are used to overlap computation and communications.

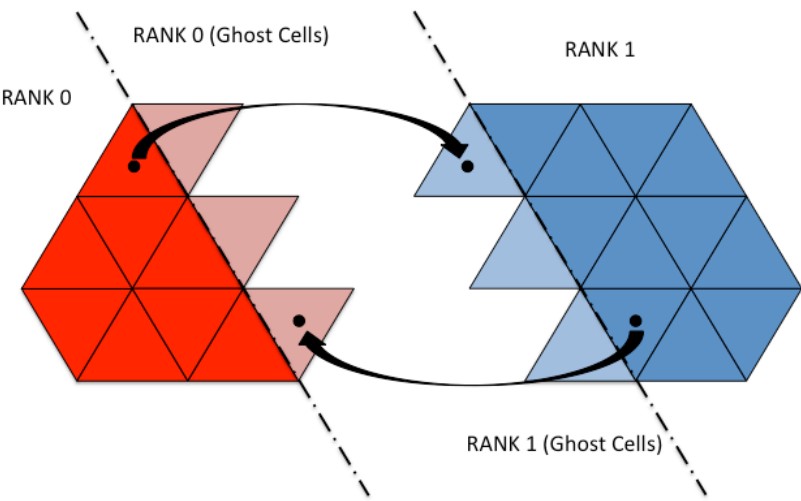

**Figure 4.** Communication pattern for the calculation of horizontal viscosity

Finally, collective communications were introduced to compute properties related to the whole domain, for instance to calculate the minimum or maximum temperature of the basin, or to calculate the total water volume. Algorithm (2) reports the

pseudo-code of the parallel implementation of the SHYFEM model.





---

**Algorithm 2** SHYFEM-MPI Time loop

---

**Require:** $\mathcal{N}$ {Set of neighbouring processes}

$\mathcal{U}, \mathcal{V}$ {Set of nodes and edges defined in the current subdomain}

**for all** $p_i \in \mathcal{N}$ **do**

    SendHalo($\mathcal{U}, \mathcal{V}, p_i$)

**end for**

**while** $t <$ num_timesteps **do**

    **for all** $p_i \in \mathcal{N}$ **do**

        RecvHalo($\mathcal{U}, \mathcal{V}, p_i$)

    **end for**

    SetExplicitTerms

    AdvanceMomentum              {eq. (1)}

    GlobalExchange(RHS)

    SolveBarotropicEquation     {eq. (4)}

    FinaliseMomentum           {eq. (5)}

    **for all** $p_i \in \mathcal{N}$ **do**

        SendHalo($\mathcal{U}, \mathcal{V}, p_i$)

    **end for**

    CalcVerticalVelocity        {eq. (6)}

    SolveTracerAdvection       {eq. (7)}

    UpdateDensity              {eq. (10)}

**end while**

---

### 4.3 Semi-implicit method for Free Surface Equation

Semi-implicit schemes are common in CFD mainly due to the numerical stability of the solution. In the case of ocean numerical modeling, external gravity waves are the fastest process and propagate at a speed of up to 200 m/s, which puts a strong constraint on the model time step in order to abide by the Courant-Friedrichs-Levy (CFL) condition for the convergence of the solution.

The semi-implicit treatment of barotropic pressure gradient, described in section (2.1), involves the solution of the matrix system

$$\mathbf{A}\eta^{n+1} = \mathbf{B} \tag{16}$$

where $\mathbf{A}$ is the matrix of coefficients that arise from the FEM discretization of derivatives of the left-hand side of eq. (4), with size $nkn \times nkn$ and $\mathbf{B}$ is the vector of the right-hand side of eq. (4). The matrix A is non-singular with irregular sparse

structure. We used PETSc (Balay et al., 1997),(Balay et al., 2020) to solve this equation efficiently.





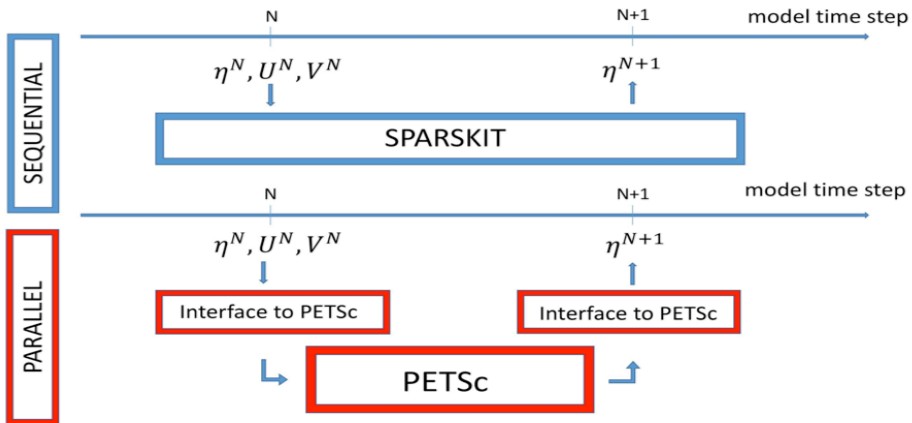

**Figure 5.** Interface to PETSc Library.

Iterative methods are the most convenient methods to solve a large sparse system with A not having particular structure properties since the direct inversion of **A** would be much too expensive. These methods search for an approximate solution for eq. (16) and include Jacobi,Gauss-Seidel, Successive Over Relaxation (SOR) and Krylov Subspace Methods (KSP). KSP is considered as one of the most important classes of numerical methods.

Algorithms based on KSP search for an approximate solution in the space generated by the matrix **A**

$$\varkappa_m(A, v_0) = span\{v_0, Av_0, A^2 v_0, \ldots, A^{m-1} v_0\} \tag{17}$$

called $m$th order Krylov subspace, where $v_0$ is an arbitrary vector (generally the right-hand side of the system) with the property that the approximate solution $x_m$ belongs to this subspace. In the iterative methods based on KSP, the subspace $\varkappa_m(A, v_0)$ is enlarged a finite number of times ($m$), where $x_m$ represent an acceptable approximate solution, giving a residual

$r_m = \mathbf{B} - \mathbf{A} x_m$ that has a smaller norm than a certain tolerance.

In a parallel application, each of the $m$ iterations is marked out by a computational cost and a communication cost to calculate the matrix-vector products in $\varkappa_m(A, v_0)$, since both the matrix and the vector are distributed across the processes. A further cost is the convergence test, which is generally based on the Euclidean norm of the residual.

$$\text{convergence test} = \begin{cases} |r_m| < a_{\text{tol}}, & \text{using absolute tolerance} \\ |r_m|/|B| < r_{\text{tol}}, & \text{using relative tolerance} \end{cases} \tag{18}$$

The calculation of the norm involves global communication to enable all the processes to have the same norm value . Hence both point-to-point and global communication burden each iteration, leading to a loss of efficiency for the parallel application if the number of necessary iterations is high. The number of iterations depends on the physical problem and on its size. An





estimate of the problem complexity is given by the condition number $C(\mathbf{A})$ which, for real symmetric matrices, is the ratio $\max(\lambda)/\min(\lambda)$ between the maximum and minimum of the eigenvalues $\lambda$ of $\mathbf{A}$. In general, the higher $C$ the more iterations

are necessary. The number of iterations also depends on the tolerance desired.

In the case of complex systems it is convenient to modify the original linear system defined in eq. (16) to get a better Krylov subspace using a further matrix $\mathbf{M}$, called the preconditioner, to search for an approximate solution in the modified system

$$\mathbf{M}^{-1}\mathbf{A}x = \mathbf{M}^{-1}\mathbf{B} \tag{19}$$

where $\mathbf{M}^{-1} \approx \mathbf{A}^{-1}$ and is computed easily.

We used PETSc rather than implementing an internal parallel solver. In fact, PETSc was developed specifically to solve problems that arise from partial differential equations on parallel architectures, and provides a wide variety of solver/preconditioners that can be switched through a namelist. In addition, the interface to PETSc is independent of the version and its implementation is highly portable on heterogeneous architectures.

Figure (5) shows the interfaces to the external packages for the native sequential version of the program (SHYFEM) and
SHYFEM-MPI.

The PETSc interface creates counterparts of $\mathbf{A}$ and $\mathbf{B}$ as objects of the package. $\mathbf{A}$ is created as a sparse matrix in coordinate format (row,column,value) using the global ID (see fig. 6) of the nodes. This is in order to have the same non-zero pattern as the sequential case, regardless of the way the domain is decomposed. The same global ID is used to build the right-hand side $\mathbf{B}$.

To solve the free surface in SHYFEM-MPI, we used the Bi-Conjucate Gradient Stabilized Method (BiCGs) with incomplete LU (iLU) factorization as a preconditioner. The convergence is reached when either the absolute tolerance ($10^{-18}$) or relative tolerance ($10^{-15}$) is reached.

The PETSc library uses a parallel algorithm to solve the linear equations. The decomposition used inside PETSc is different from the domain decomposition used in SHYFEM-MPI (see fig. 6). PETSc divides the matrix $\mathbf{A}$ in ascending order with the
global ID. The SHYFEM-MPI partitioning is based on criteria that take into account the geometry of the mesh and is, at any rate, different from PETSc. For this reason, after the approximate solution for $\eta^{n+1}$ has been found by PETSc, the solution vector is gathered by the master process and is redistributed across the MPI processes.

### 4.4  I/O Management

I/O management usually represents a bottleneck in a parallel application. To avoid this, input and output files should be
concurrently accessed by the parallel processes, and each process should load its own data. However, loading the whole file for each process would affect memory scalability. In fact, the allocated memory should be independent of the number of parallel processes in order to ensure the memory scalability of the code. The two issues can be addressed by distributing the I/O operations among the parallel processes. During the initialization phase, SHYFEM needs to read two files: the basin geometry and the namelist. All the MPI processes perform the same operation and store common information. This phase is



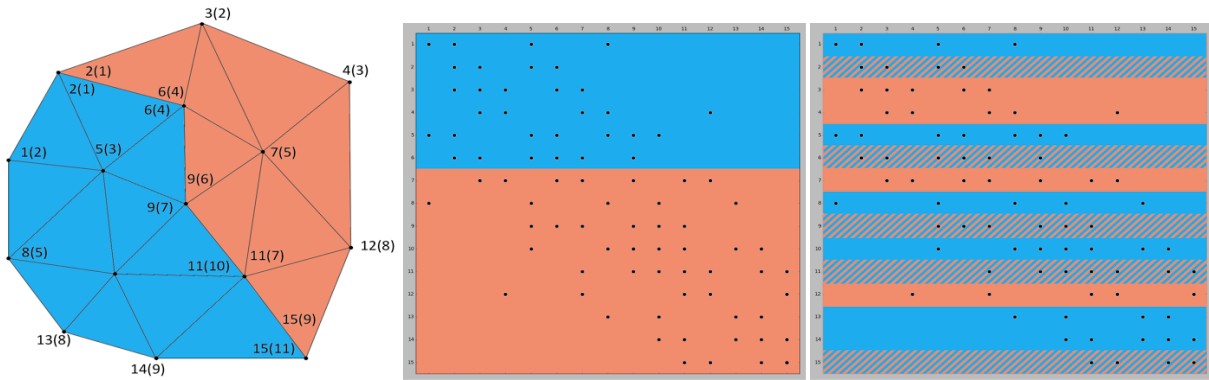

**Figure 6. [top]** Example of partition in 2 MPI processes - Process 0 (Blue) and Process 1 (Red). Numbers represent the Global (Local) ID **[bottom left]** Partition of matrix A in PETSc - Rows in the matrix are sequentially allocated to the processes **[bottom right]** Partition of matrix A in SHYFEM-MPI - banded colors indicate rows of the matrix corresponding to shared nodes

not scalable because each process browses the files. However, this operation is only performed once and has a limited impact on the total execution time. As a second step, initial conditions and forcing (both lateral and surface) are accessed by all the parallel processes, but each one reads its own portion of data, as shown in Figure 7. Surface atmospheric forcing is defined on a structured grid, so after reading the forcing file, each process interpolates the data on the unstructured grid used by the model. The output can be written using external parallel libraries capable of handling parallel I/O. A check-pointing mechanism was

implemented in the parallel version of the model. This is usual when the simulation is divided into dependent chunks in order to maintain the status. Both phases (reading/writing) are performed in a distributed way among the processes to reduce the impact on the parallel speedup of the model, as shown in Figure 7. Each process generates its own restart file related to its sub-domain and reads its own restart file.

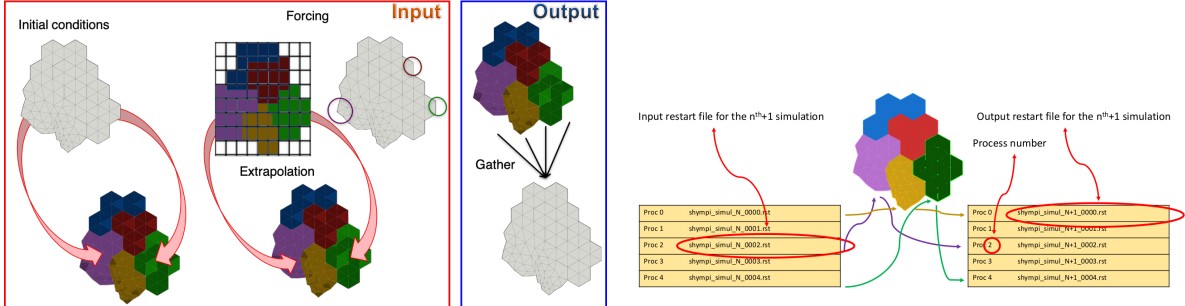

**Figure 7.** Data management related to the forcing files and restart files.





## 5 Results

We ran our experiments to assess the correctness of MPI implementation on the Southern Adriatic Northern Ionian coastal Forecasting System (SANIFS) configuration, which has a horizontal resolution of 500m near the coast of up to 3-4km in open waters. The total number of grid elements is 176,331. Vertical resolution is 2m near the surface stepwise increasing towards the sea bottom, dividing the water column into 80 layers. For details of the model grid and the system settings see (Federico et al., 2017).

Runs are initialized with the motionless velocity field and with temperature and salinity fields from CMEMS NRT products (Clementi et al., 2019). The simulations are forced hourly at lateral boundaries with water level fields, total velocities and active tracers from the same products.

The sea level is imposed with a Dirichlet condition, while relaxation is applied to the parent model total velocities with a relaxation time of 1 hour. For scalars, the inner values are advected outside the domain when the flow is outwards, while a
Dirichlet condition is applied for inflows.

The boundary conditions for the upper surface follow the MFS bulk formulation (Pettenuzzo et al., 2010), which requires wind, cloud cover, air and dew point temperature, available in ECMWF analysis, with a temporal/spatial resolution of 6 hours/0.125 degrees respectively. From the same analysis, we force the surface with precipitation data.

We select an upwind scheme for both horizontal and vertical tracer advections. The formulation of bottom stress is quadratic.
The time stepping for the hydrodynamics is semi-implicit with $\gamma = \beta = 0.6$. Horizontal viscosity / diffusivity follows the formulation of (Smagorinsky, 1963). Turbulent viscosity / diffusivity is set to a constant value equal to $10^{-3} \mathrm{m}^2/\mathrm{s}$.

The cold start implies strong baroclinic gradients. To prevent instabilities, we select a relatively small time step, set to $\Delta t = 15\mathrm{s}$.

### 5.1 SHYFEM-MPI Model validation

The parallel implementation of the SHYFEM model was validated to assess the reproducibility of the results when varying the size of the domain decomposition and the number of parallel cores used for a simulation. Our baseline was the results from a sequential run and we compared the results with those obtained with parallel simulations on [36,72,108,216] cores. The parallel architecture used for the tests is named Zeus and is available at the CMCC Supercomputing Center. Zeus is a parallel machine equipped with 348 parallel nodes interconnected with an Infiniband EDR (100Gbps) switch. Each node has two Intel
Xeon Gold 6154 (18cores) CPUs, and 96GB of main memory.

The results were compared with a one-day simulation, saving the outputs every hour (the model executes 5760 timesteps) and referring to the data from the native grid. Only the most significative fields were taken into account for comparison: temperature, salinity, sea surface high and zonal velocity. In order to evaluate the differences between the parallel execution with respect to the results obtained with a sequential run, we used the root mean square error as a metric:

$$\mathrm{RMSE}(X, Y) = \sqrt{\frac{\sum_i ((X_i - Y_i)^2)}{N}}$$



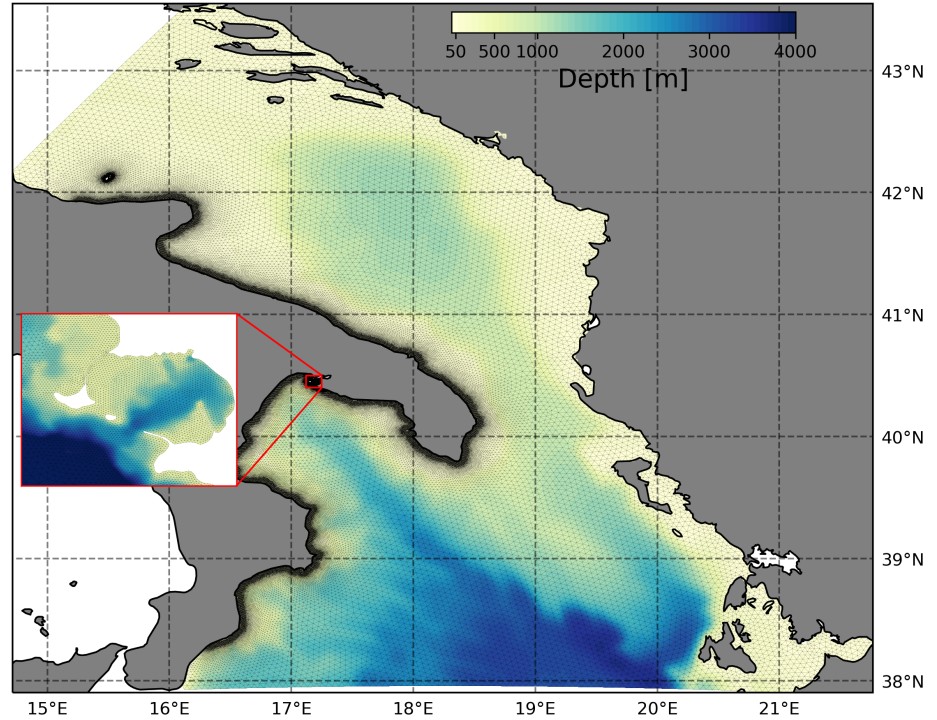

**Figure 8.** Domain of the SANIFS configuration. The model mesh is superimposed over bathymetry

As a result, we computed the RMSE for each domain decomposition, for each aforementioned field and for each timestep saved in the output files (hourly), as shown in Figure (9). The RMSE timeseries were calculated using NCO operators applied on the output on the native SHYFEM-MPI grid considering all the active cells in the domain.

The timeseries of all the MPI decompositions overlap, which means that the program can reproduce the sequential result close to machine precision. The timeseries of the SSH are noisy, but the RMSE remains steady and of the order of $10^{-13}$. The SHYFEM-MPI model is not bit-to-bit reproducible for two main reasons. The field values computed on the grid nodes belonging to the border between two domains are computed first by the two processes independently, taking into account only the node neighbours in the local domain thus giving a partial value. The partial values are then exchanged between the two processes and the final value is computed with a sum of the partial values. The order of the floating point operations, executed on the grid nodes at the border, thus changes when the domain decomposition changes, creating a numerical difference between two simulations that use a different number of cores. In fact, the floating-point operations lose their associative property due to the approximate representation of the numbers (Goldberg, 1991). The second source of non bit-to-bit reproducibility is due to the optimized implementation of the PETSc numerical library which makes use of non-blocking MPI communications. This, in turn, creates a non-deterministic order in the execution of the floating-point operations, which generates a numerical difference



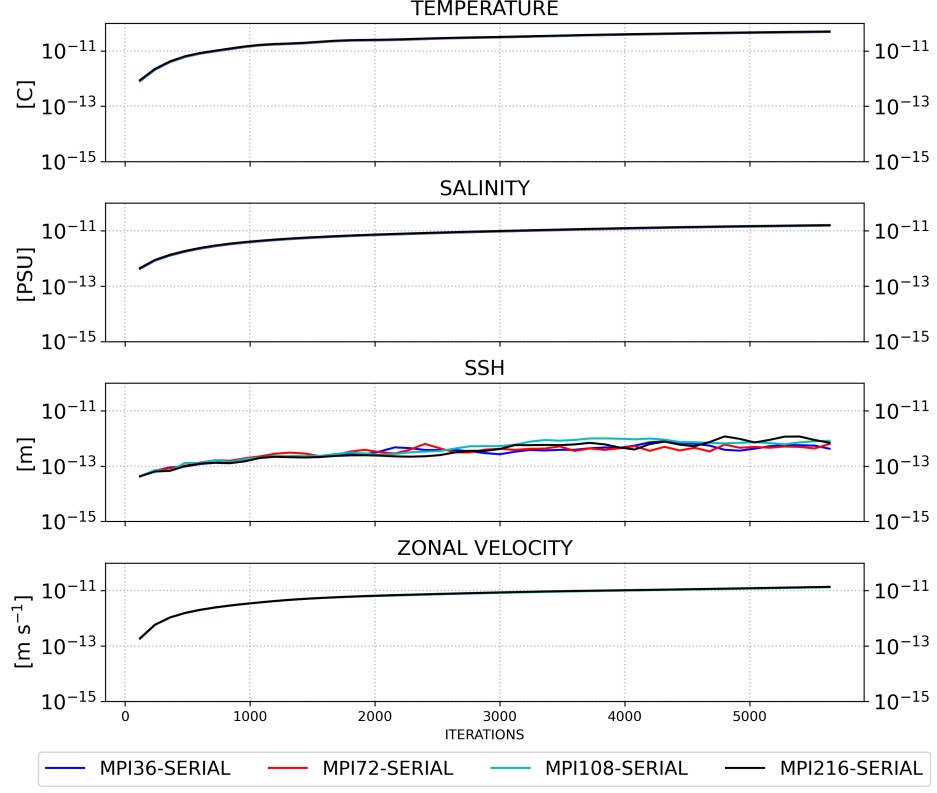

**Figure 9.** RMSE timeseries of SHYFEM-MPI outputs compared with the sequential run for all the prognostic fields and with different numbers of cores: 36, 72, 108 and 216.

even between two different executions of the same configuration with the same domain decomposition and the same number of cores.

To further assess the non-deterministic behaviour of the PETSc solver, we ran the same configuration five times with the same number of cores. Again we used the RMSE as a metric to quantify the differences between four simulations with respect to the first one which was taken as reference. Figure (10) shows the RMSE timeseries of the simulations executed with 72 cores.

Although the SHYFEM-MPI implementation is not bit-to-bit reproducible, the RMSE timeseries show that for each of the model variables, the deviations of the runs from the reference run remain close to the machine precision, with no effect on the reproducibility of the solution of the physical problem. We implemented a perfect reproducible version of the model by including a halo over the elements shared between the neighbouring processes. This ensures that the order of the floating-point operations is kept the same and forces the PETSc solver for sequential use. However, this version is only for debugging and is beyond the scope of this work.



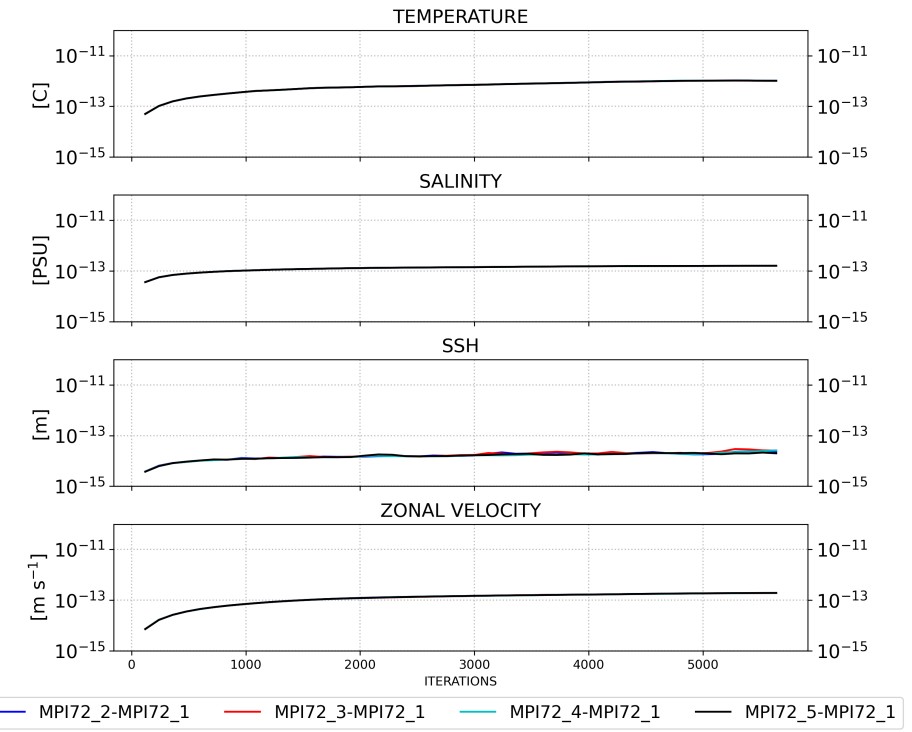

**Figure 10.** RMSE evaluated for code reproducibility with 72 MPI processes

## 5.2 SHYFEM-MPI Performance assessment

The parallel scalability of the SHYFEM-MPI model was evaluated on Zeus parallel architecture. The SANIFS configuration was simulated for seven days and the number of cores varied up to 288. A total of 36 core multiples were used since a Zeus computing node is equipped with two sockets of 18-cores each, hence 36 cores were available.

Figure (11) shows the total execution time in a log-log plot where the parallel scalability can also be evaluated. In fact, the behaviour of an ideal scalability results in a straight line in the log-log plot. The labels associated with each point in the plot represent the parallel efficiency. The efficiency drops below 40% with 288 cores, which can thus be considered as the scalability bound of the model for the SANIFS configuration.

Deeper insights into the performances are provided in figures (12) which report the execution time for different processing steps of the model. The execution time was partitioned between the routines to evaluate the momentum equation, advection/diffusion equation, sea level equation (which involves the PETSc solver), the MPI communication time, and the I/O. The results demonstrate a very good scalability for the momentum and trace processing steps. The limiting factors to the overall scalability lie in the sea level computation which involves the PETSc solver and in the overhead introduced by the MPI communications. The communication time also includes the idle time required for waiting for the slowest process to reach the communication call. The idle time can be reduced by enhancing the work load balance among the processes. Although the

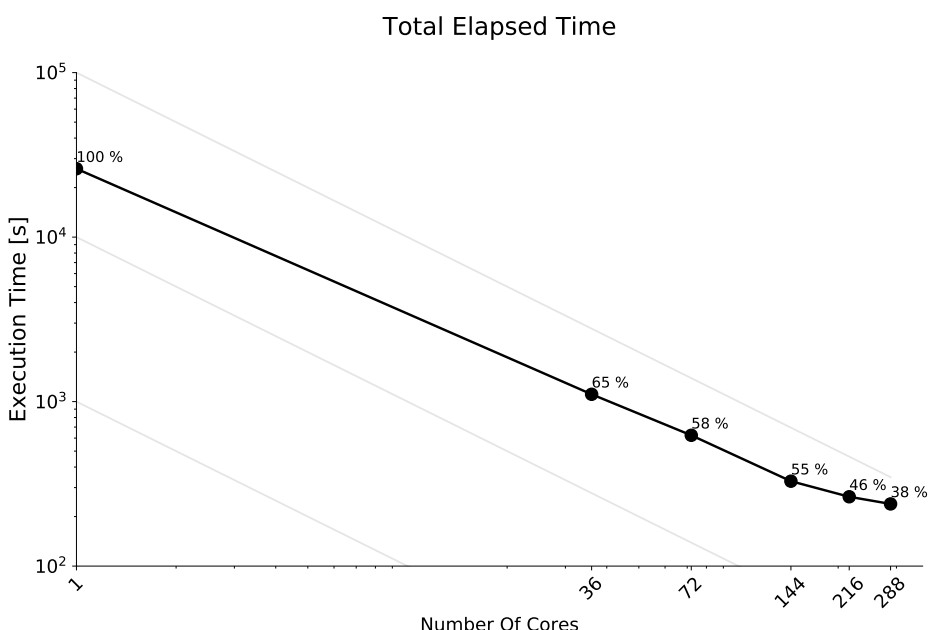

**Figure 11.** SANIFS execution time in a log-log plot

time spent for I/O is completely unscalable, it is not a limiting factor in this experiment since it is two orders of magnitude smaller than the other processing steps.

Figure (13) shows the ratio in the execution time of the model's processing steps. The evaluation of the momentum equation

and the advection/diffusion equation take most of the execution time in the sequential run. Increasing the MPI processes, the ratios among these components change, and the communication cost becomes an increasing burden on the total execution time.

To conclude, the free surface equation part of the code needs to be better investigated for a more efficient parallelization. The execution time reported for the free surface solver includes the assembly of the linear system, the communication time of the internal routines of PETSc for each of the solver iterations, and the communication needed to redistribute the solution onto

the model grid. The effects of a non-optimal model mesh partitioning on the solver efficiency have not yet been assessed, nor has whether the combination preconditioner/solver iLU+BiCgS is the most suitable in the context of parallel performances. Moreover, a more efficient partition algorithm should be adopted to reduce the idle time and to improve the load balancing.

## 6   Conclusions

The hydrodynamical core of SHYFEM is parallelized with a distributed memory strategy, allowing for both calculation and

memory scalability. The implementation of the parallel version includes external libraries for domain partitioning and the solution of the free surface equation. The parallel code was validated using a realistic configuration as a benchmark. The optimized version of the parallel model does not reproduce the output of the sequential code bit-to-bit, but reproduces the



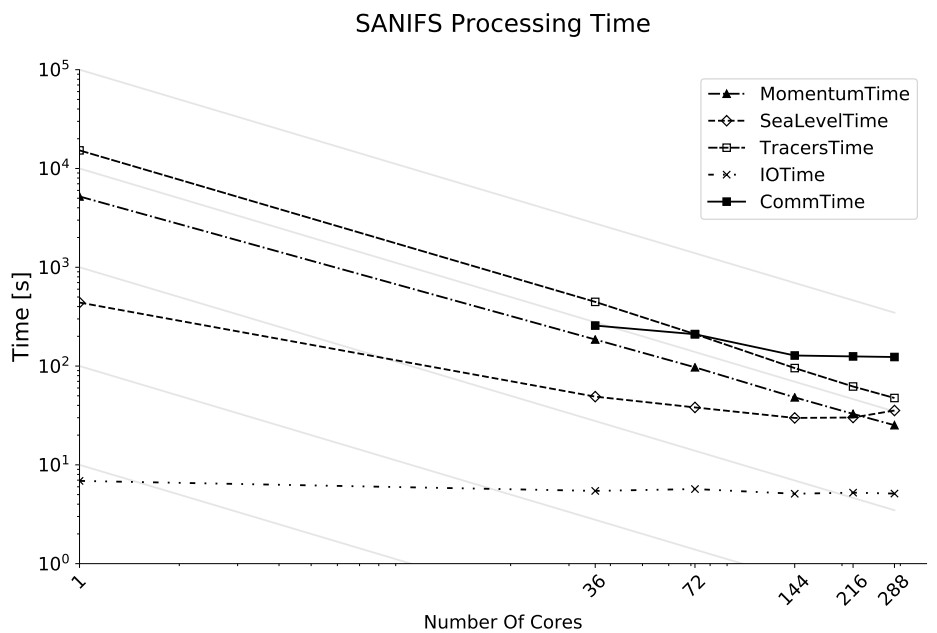

**Figure 12.** SANIFS detailed execution time for different processing phases

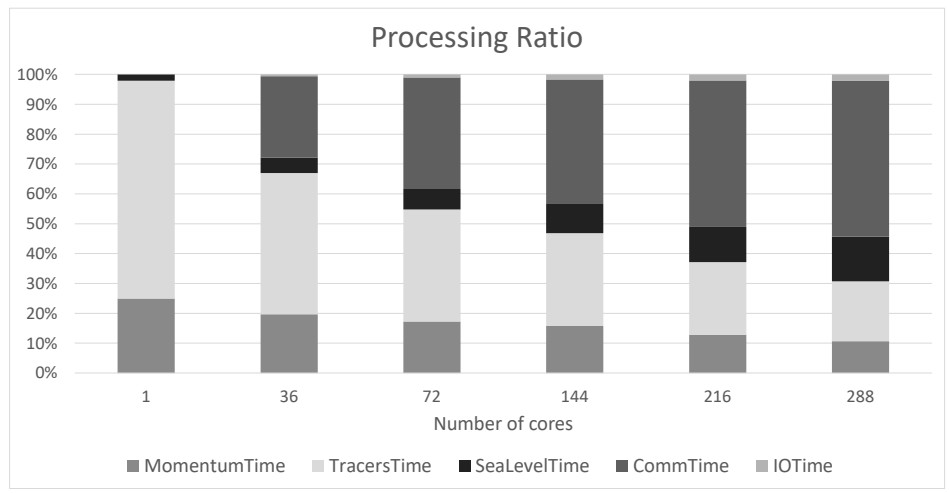

**Figure 13.** SANIFS processing ratio among the computing phases

physics of the problem without significant differences with respect to the sequential run. The source of these differences was considered for different orders of operations in each of the domain decompositions. Forcing the code to exactly reproduce the order of the operation in the sequential code was found to lead to a dramatic loss of efficiency, and was therefore not considered in this work.





Our assessment reveals that the limit of scalability in the parallel code is reached at 288 MPI cores, when the parallel efficiency drops below 40%. The analysis of the parallel performance indicates that with a high level of MPI processes used, the burden of communication and the cost of solving the free surface equation take up a huge proportion of the single model time step. The workload balance needs to be improved, with a more suitable solution for domain partitioning. The parallel code, however, enables one of the main tasks of this work to be accomplished, namely to obtain the results of the simulation in a time that is reasonable and significantly faster than the sequential case. The benchmark has demonstrated that the execution time is reduced from nearly eight hours for the sequential run to less than four minutes with 288 MPI cores.

*Code availability.* The code of the parallel version of the SHYFEM is available at https://doi.org/10.5281/zenodo.5596734 and it is citable with DOI: 10.5281/zenodo.5596734

## Appendix A: Linear Shapefunctions

A continuous function $f(x,y)$ in the 2D space can be represented onto a discrete mesh as a linear combination of base functions $\phi_k$

$$f(x,y) \approx \sum_k \phi_k(x,y) f_k \tag{A1}$$

where $f_k$ denote the coefficients of the functions that approximate $f$. In the context of SHYFEM, $\phi$ functions are node-referenced linear functions that overlap only in elements in common between adjacent nodes. In particular, the shapefunction of a node k is 1 on k and 0 on the others (fig. A1)

Considering an element e with its three nodes $(k_1,k_2,k_3)$ of coordinates $(x_1,y_1),(x_2,y_2),(x_3,y_3)$ with the corresponding values $(f_1,f_2,f_3)$ of $f$ we are interested in the gradient of $f$ within the element e. We consider the shapefunctions that overlap in this element as $\phi_{k_1,e},\phi_{k_2,e},\phi_{k_3,e}$ (see fig. A2) with the following constraints:

$$f(x,y) = f_1 \phi_{k_1,e} + f_2 \phi_{k_2,e} + f_3 \phi_{k_3,e} \tag{A1}$$

$$\frac{\partial f}{\partial x} = f_1 \frac{\partial \phi_{k_1,e}}{\partial x} + f_2 \frac{\partial \phi_{k2,e}}{\partial x} + f_3 \frac{\partial \phi_{k3,e}}{\partial x} \tag{A2}$$

$$\frac{\partial f}{\partial y} = f_1 \frac{\partial \phi_{k_1,e}}{\partial y} + f_2 \frac{\partial \phi_{k_2,e}}{\partial y} + f_3 \frac{\partial \phi_{k_3,e}}{\partial y} \tag{A3}$$

The shapefunctions satisfy the following relations:





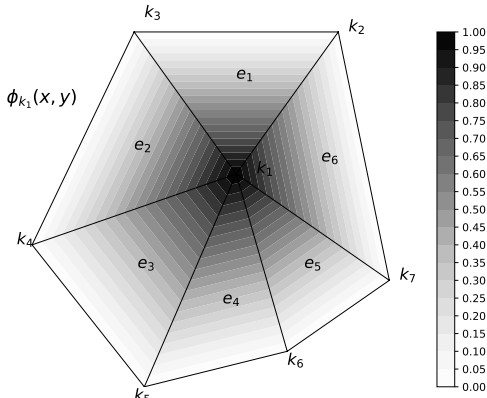

**Figure A1.** Shapefunction of node $k_1$. The function is 1 in $k_1$ and 0 in all the other nodes. $\phi_{k_1}(x, y)$ overlaps only with the shapefunctions associated with neighbouring nodes.

$$\phi_{k_1,e} + \phi_{k_2,e} + \phi_{k_3,e} = 1 \tag{A4}$$

$$\phi_{k_1,e}x_1 + \phi_{k_2,e}x_2 + \phi_{k3,e}x_3 = x \tag{A5}$$

$$\phi_{k_1,e}y_1 + \phi_{k_2,e}y_2 + \phi_{k3,e}y_3 = y \tag{A6}$$

which can be written in matrix form:

$$\begin{bmatrix} 1 & 1 & 1 \\ x_1 & x_2 & x_3 \\ y_1 & y_2 & y_3 \end{bmatrix} \begin{bmatrix} \phi_{k1,e} \\ \phi_{k2,e} \\ \phi_{k3,e} \end{bmatrix} = \begin{bmatrix} 1 \\ x \\ y \end{bmatrix} \tag{A7}$$

or, in compact form

$$A \begin{bmatrix} \phi_{k1,e} \\ \phi_{k2,e} \\ \phi_{k3,e} \end{bmatrix} = \begin{bmatrix} 1 \\ x \\ y \end{bmatrix} \tag{A8}$$

The shape functions are calculated by inverting the system (A8). Here we report the expression of shapefunctions of the k nodes in the element e (fig. A2)





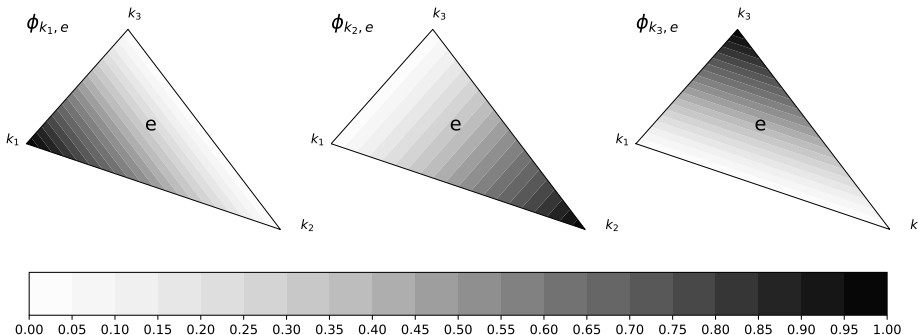

**Figure A2.** Shapefunctions overlapping in the element $e$

$$\phi_{k_1,e} = \frac{1}{|A|}[(x_2 y_3 - y_2 x_3) + (y_2 - y_3)x + (x_3 - x_2)y] \tag{A9}$$

$$\phi_{k_2,e} = \frac{1}{|A|}[(x_3 y_1 - x_1 y_3) + (y_3 - y_1)x + (x_1 - x_3)y] \tag{A10}$$

$$\phi_{k_3,e} = \frac{1}{|A|}[(x_1 y_2 - x_2 y_1) + (y_1 - y_2)x + (x_2 - x_1)y] \tag{A11}$$

and their derivatives

$$\frac{\partial \phi_{k_1,e}}{\partial x} = \frac{y_2 - y_3}{|A|}, \frac{\partial \phi_{k_1,e}}{\partial y} = \frac{x_3 - x_2}{|A|} \tag{A12}$$

$$\frac{\partial \phi_{k_2,e}}{\partial x} = \frac{y_3 - y_1}{|A|}, \frac{\partial \phi_{k_2,e}}{\partial y} = \frac{x_1 - x_3}{|A|} \tag{A13}$$

$$\frac{\partial \phi_{k_3,e}}{\partial x} = \frac{y_1 - y_2}{|A|}, \frac{\partial \phi_{k_3,e}}{\partial y} = \frac{x_2 - x_1}{|A|} \tag{A14}$$

## Appendix B: Vertical Viscosity and Advection Operators

The integration of vertical viscosity term $\frac{\partial}{\partial z}\left(\nu \frac{\partial \boldsymbol{u}}{\partial z}\right)$ in the momentum equation over a generic layer $l$ as in figure (C1) leads to

$$\int_{z_l}^{z_{l-1}} \frac{\partial}{\partial z}\left(\nu \frac{\partial \boldsymbol{u}}{\partial z}\right) \mathrm{d}z = \nu \frac{\partial \boldsymbol{u}}{\partial z}\bigg|_{z_{l-1}} - \nu \frac{\partial \boldsymbol{u}}{\partial z}\bigg|_{z_l} \tag{B1}$$

The stresses are discretized with centered differences

$$\nu \frac{\partial \boldsymbol{u}}{\partial z}\bigg|_{z_{l-1}} - \nu \frac{\partial \boldsymbol{u}}{\partial z}\bigg|_{z_l} = \nu_{l-1} \frac{\boldsymbol{u}_{l-1} - \boldsymbol{u}_l}{(h_{l-1} + h_l)/2} - \nu_l \frac{\boldsymbol{u}_l - \boldsymbol{u}_{l+1}}{(h_l + h_{l+1})/2} \tag{B2}$$





**Figure C1.** Distribution of variables in 3 generic layers $l-1, l, l+1$

Grouping the velocities by layer and using the identity $\boldsymbol{U}_l = \boldsymbol{u}_l h_l$ we write the difference of stresses as a vertical viscosity operator $D_z$ applied to the velocity integrated in the layer $l$ appearing in eq. (1)

$$
\nu \frac{\partial \boldsymbol{u}}{\partial z}\bigg|_{z_{l-1}} - \nu \frac{\partial \boldsymbol{u}}{\partial z}\bigg|_{z_l} = \left( \frac{\nu_{l-1}}{h_{l-1}(h_{l-1}+h_l)/2} \boldsymbol{U}_{l-1} \right.
$$
$$
- \frac{1}{h_l}\left[ \frac{\nu_{l-1}}{(h_{l-1}+h_l)/2} + \frac{\nu_l}{(h_l+h_{l+1})/2} \right] \boldsymbol{U}_l \tag{B3}
$$
$$
\left. + \frac{\nu_l}{h_{l+1}(h_l+h_{l+1})/2} \boldsymbol{U}_{l+1} \right) \equiv D_z \boldsymbol{U}_l
$$

The vertical advection term that appears in eq. (2) is discretized under the assumption that horizontal divergence is small
$(-\nabla \boldsymbol{u} = \partial w/\partial z \approx 0)$ and hence $w$ does not vary within the layer $l$ (see fig. C1)

$$
\int_{z_l}^{z_{l-1}} w^n \frac{\partial \boldsymbol{u}^n}{\partial z} \mathrm{d}z \approx <w>_l \int_{z_l}^{z_{l-1}} \frac{\partial \boldsymbol{u}^n}{\partial z} \mathrm{d}z = <w>_l \cdot [\boldsymbol{u}^n\big|_{z_{l-1}} - \boldsymbol{u}^n\big|_{z_l}] \tag{C1}
$$

where $<w>_l$ is the average of $w$ at $z_{l-1}$ and $z_l$. The horizontal velocity is calculated on the interfaces by means of a weighted average of corresponding upper/lower layer

$$
\int_{z_l}^{z_{l-1}} w^n \frac{\partial \boldsymbol{u}^n}{\partial z} \mathrm{d}z \approx \frac{w_{l-1}^n + w_l^n}{2} \cdot \left[ \frac{\boldsymbol{u}_{l-1}^n h_{l-1} + \boldsymbol{u}_l^n h_l}{h_{l-1} + h_l} - \frac{\boldsymbol{u}_l^n h_l + \boldsymbol{u}_{l+1}^n h_{l+1}}{h_{l+1} + h_l} \right] \tag{C2}
$$





*Acknowledgements.* This work was part of the Strategic project #4 "A multihazard prediction and analysis testbed for the global coastal ocean" of CMCC.

*Author contributions.* N.Pinardi, G.Aloisio, P.Schiano and G.Coppini formulated the research goals; S.Mocavero, I. Epicoco, G. Micaletto and I. Barletta designed the parallel algorithm; G. Micaletto and S. Mocavero developed software for the parallel algorithm; I. Barletta, I. Federico and N. Pinardi validated the parallel model and conducted the formal analysis; G. Micaletto and I. Barletta performed the 500 computational experiments; G. Aloisio, P. Schiano and N. Pinardi provided the computational resources; G. Micaletto, S. Mocavero and I. Barletta wrote the first draft of the manuscript; I. Barletta and I. Federico provided the data visualization; G. Micaletto, S. Mocavero and I.Epicoco assessed the computational performance; I.Federico, G. Verri and I. Barletta contributed to the code tracking of the model equations; N. Pinardi, G. Aloisio, P. Schiano and G.Coppini supervised the research; all the authors contributed to writing and revising the manuscript.

*Competing interests.* No competing interests are present



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
