# Peer review of "Parallel Implementation of the SHYFEM Model"

_Geoscientific Model Development, 2021_

## Author Response (AR1)

**SUMMARY OF CHANGES**

GIORGIO MICALETTO, IVANO BARLETTA, SILVIA MOCAVERO, IVAN FEDERICO, ITALO
EPICOCO, GIORGIA VERRI, GIOVANNI COPPINI, PASQUALE SCHIANO, GIOVANNI ALOISIO,
AND NADIA PINARDI

To the Topical Editors of Geoscientific Model Development

Dear Topical Editor,

The revised version of the manuscript GMD-2021-319 entitled "Parallel Implementation of the SHYFEM Model" has been edited to meet the suggestions included in the revision.

**1. REVIEW #1**

*As claimed by the authors the limiting factors to the overall scalability lie in the sea level computation which involves the PETSc – KSP solver and data decomposition. The efficiency drops below 55% with 144 cores. In particular, Matrix decomposition used in PETSc, namely the block row partition, is not the same of SHYFEM. Then, a global communication is required and a loss of efficiency results. My questions: are since the parallelism is being introduced in the original sequential version of SHYFEM, why the data partitioning chosen for SHYFEM is not the same of PETSC?*

**Summary of changes:** The grid points (nodes) are ordered after application of the Cuthill-McKee algorithm on the mesh. This approach is used in continuity with the sequential version of SHYFEM, where the Cuthill-McKee algorithm is embedded in the SPARSEKIT framework. This is done to optimize the bandwidth of the sea level matrix and make the iterative solution faster. PETSc uses a 2D domain decomposition based on the ordering of the nodes in the matrix without considering the geometry; whilest, the grid partitioning used in SHYFEM takes into account the load balancing for the computation made on the 3D domain and to minimize the spatial cutting edge between one process and its neighbors (regardless of the numbering of the elements in the matrix). Moreover, the decomposition in SHYFEM is done considering the elements and not nodes as in PETSc. The decomposition used in SHYFEM is optimal to reduce the communications overhead and to better balance the workload in all other aspects of the model which computationally represent the most onerous part.

*PETSc offers the Distributed Arrays (DMDA) objects that simplify the distribution and the management of the domain data (all the physical quantities on the domain region) in*

*a distributed memory system. Why the authors do not exploit DMDA objects?*

**Summary of changes:** To our knowledge, the DMDA objects are exclusively oriented to structured grids, as stated from PETSc manual: "structured grid in 1, 2, or 3 dimensions. In the global representation of the vector each process stores a non-overlapping rectangular". In our case the SHYFEM model uses an unstructured grid.

*Otherwise, why do not explore the use of TRILINOS, or HYPRE that are already able to interact with PETSc ?*

**Summary of changes:** We thank the reviewer for pointing out this aspect. The exploitation of Trilinos and Hypre libraries is under evaluation and we plan to include them in the code as soon as we have evidence of an improvement of the computational performance w.r.t. the current implementation, unfortunately this activity takes much more time than expected and cannot be considered for this manuscript. Even if the current implementation is perfectible, we believe we have reached a milestone by parallelizing the SHYFEM model with MPI and introducing the use of high performance numerical libraries.

*Finally, why the authors do not explore matrix-free solvers?*

**Summary of changes:** We thank the reviewer for this suggestion, we have implemented a matrix-free approach and compared the computational performance w.r.t. the current implementation. The new approach did not bring an evident improvement, we reported this analysis in Section 5.2 of the paper. In particular, the matrix free approach requires the user to explicitly write the routine for the matrix-vector multiplication, however in order to correctly compute the matrix-vector multiplication a communication stage among the processes is needed. This communication time partially invalidates the benefit given by the matrix-free approach.

*The KSP solver used for the free surface equation is known to have synchronization points at each iteration leading to a loss of efficiency for the parallel algorithm. My question is: why do not explore communication avoiding variants of Krylov sparse solvers?*

**Summary of changes:** We tested some other solvers which exploit the communication avoiding technique. This further analysis is reported in Section 5.2. We evaluated the following solvers: Generalized Minimal Residual(GMRES); Improved Biconjugate gradient stabilized method(IBCGS), Flexible Biconjugate gradient stabilized (FBCGSR) and Biconjugate gradient stabilized method(BCGS). Among these methods the most efficient one, in our configuration, was the FBCGSR. We further investigated two pipelined methods such as PIPEBCGS and PIPEFGMRES: the first method diverges after a few iterations, it was necessary to increase the tolerances by 4 orders of magnitude in order to use it and it did not lead to improvements, the second instead had a worse scalability than the BCGS method used initially.

*In addition to these two factors, in my opinion there is another crucial point. Since domain decomposition involves only spatial direction and not the time direction, a global communication is required at each time step of surface equation. My question is:*
*why do not explore parallel -in -time approaches ? Parallelism should be introduced ab initio in any mathematical / numerical model, and this is especially true for time marching models. Otherwise, the efficiency will be ever poor.*

**Summary of changes:** We thank the reviewer for pointing out this aspect. The parallel in time approach would require a complete and disruptive revision of the data structure and the whole code structure. The parallel in time approach is out of the scope of the presented work and it can be considered as a future investigation.

*In conclusion, I think that while the deployment of an application software by means of the use of scientific libraries, such as PETSc, can be considered a good investment, SHYFEM needs to be deeply redesigned to meet scalability requirements.*

**Summary of changes:** We are aware that the code can be further optimized in the future. Our work followed an evolutionary approach starting from an already existing sequential implementation of the model. A deep optimization would require a complete redesign of the code structure but this was out of the scope of our work.

**2. Review #2**

*- please define graph partitioning with one sentence in section 3.3 for the readers that are not familiar with the concept.*

**Summary of changes:** We inserted the following sentence in section 3.3 to better explain the concept of graph partitioning: "graph partition consists in the aggregation of nodes into mutually exclusive groups minimizing the number of cutting edge. The graph partitioning problems fall under the category of NP-hard problems. Solutions to these problems are generally derived using heuristics and approximation algorithms".

*- In figure 5, it is demonstrated that the sequential code uses SPARSEKIT. Please add a short description of it to the text. Is this the tool mentioned in the following link, https://www-users.cse.umn.edu/ saad/software/SPARSKIT/. If so, please also include it in the references. If it is not relevant to include it, then the sequential part can be removed from the figure.*

**Summary of changes:** We removed figure 5 at all because it did not add essential information to the text.

*- the authors are discussing the convergence criteria In Section 4.3, specifically at line 326. Is there any other criteria used for the convergence such as maximum number of iteration etc.? In some cases/applications, it would be hard to meet the defined tolerance criteria.*

**Summary of changes:** We modified the text in Section 4.3 reporting the four convergence criteria of KSP method:
- the residual is lower than the tolerance
- the relative residual is lower than the relative tolerance
- residual is larger than divergence tolerance
- the number of iterations exceeds the maximum number of iterations
Moreover, we added the following sentence after equation (18): "The last two criteria possibly indicate that the method is diverging and that the solution is not accurate" and finally, will replaced line 326 with: "We set absolute and relative tolerance to $10^{-12}$ and $10^{-15}$ respectively. Divergence tolerance and maximum iterations are set to default values (both $10^4$)".

*- It seems that the solution vector needs to be gathered and redistributed again due to the mismatch between the domain composition defined in the model and the PETSc. As it indicated in the manuscript, the domain decomposition is defined in the pre-processing stage at the beginning of the simulation through the use of the Zoltan package. Is it possible to reorder decomposition in the PETSc to get rid of collecting and redistributing the solution in every iteration? It could create a bottleneck in the overall performance of the model especially for the high resolution applications. The performance benchmark results are also supporting low efficiency of the model when the number of processors is increased. Is there any available tool that creates a link between Zoltan and PETSc to solve similar issues? If the authors could clarify it in the text that would be great. It would be also nice to add references that mention about the similar issue.*

**Summary of changes:** We thank the reviewer for pointing out this aspect. To the best of our knowledge, no tool exists that creates a link between Zoltan and PETSc. However, we are investigating the exploitation of the DMPlex module within PETSc to handle unstructured model meshes. The FESOM model is one example of an unstructured model which uses PETSc for solving the free surface equation.

*- The section 4.4 mainly aims to give information about the I/O management but it does not give details of the parallel I/O library used to read initial and forcing data. This section needs to be extended to include the details of the used library for the parallel I/O and also its scalability (may be in the validation section). Additionally, since each process reads its own restart file, it seems that it is not possible to restart the model using a different number of processors than used in the initial run. This might add extra limitations to the usability of the model and need to be more emphasized. Since the model restarted in a parallel fashion, is there any issue related with the bit-to-bit reproducibility of the simulation*

*results. What are the restrictions in the current implementation in terms of reproducibility?*

**Summary of changes:** The input files are mainly represented by binary unformatted files; the parallel I/O is directly implemented in the code. We confirm that in this version of the model the restart files are read by each process and it is not possible to change the number of processes between one run and the next one. However, we are extending the code to support the restarting from a single restart file with the global domain data. The reproducibility analysis has been explained in section 5.1 and the factors that impact on the bit-to-bit reproducibility are due to different order in which the floating point operations are evaluated.

The restartability of the model has been validated comparing the outputs obtained by a "LONG" run simulating 2 days but after the first day of simulation the restart files are saved, and the "SHORT" run which simulates only one day restarting from the restart files saved by the "LONG" run. We obtained a bit-to-bit comparison between the outputs obtained from the LONG run and those obtained from the SHORT run. For the restartability test we used the version of the model compiled in DEBUGON mode (which build a bit-to-bit reproducible executable). We will add this explanation in section 5.1 of the manuscript.

*- Which tool is used to interpolate the forcing data from the regular grid to the unstructured mesh? Is it custom code? What kind of interpolation algorithm is used here (conservative, bilinear etc.). Since the forcing might include flux components, it would be nice to use conservative type interpolation for those variables. It would be also nice to include information about the interpolation of vector fields and how they are handled. The tool might need to preserve the properties of the vector fields after interpolation.*

**Summary of changes:** The model has its own internal routines to build weights and perform bilinear interpolation of atmospheric forcing fields at runtime. Forcing fields are also interpolated linearly in time. The aforementioned routines work also in parallel where each process applies the weights within its own portion of the domain, as described in section 4.4. In the parallel version we kept the same approach used in the sequential version of the model, the application of conservative interpolation methods goes beyond the scope of this paper, which regards the optimization of the code only from an HPC perspective.

*- To assess the decoupled effect of the I/O and computational performance of the model, it would be better to run the model with the configuration that does not write the output. By this way, the pure performance of the MPI implementation can be clearly seen. Since, I/O can be affected by its way of implementation in the model and the architecture of the underlying system (number of I/O nodes, their interconnection, number MDS and OSS servers, RAID configuration and used parallel file system) it would be hard to assess the effect of the I/O in the benchmark results and could add extra uncertainty of the performance measurements and results. Personally, I prefer to see the pure MPI communication overhead in the benchmark results.*

**Summary of changes:** The performance assessment has been executed after careful instrumentation of the code in order to measure the execution time spent within the I/O routines and within the MPI calls to evaluate the communication overhead and within the main part of the computation. Figure 12 reports the scalability analysis for the different phases of the code in a configuration which is close enough to a realistic run. However, innaccordance with the reviewer suggestion, we extended Section 5.2 with the performance evaluation when the outputs is disabled.

*- memory scaling is mentioned about one of the key advantages of the distributed memory approach but there is no any information about the scaling of the memory in the current implementation. It would be nice to add a plot or table (total memory per node vs. number of processors etc.) that includes more information about memory scaling of SHYFEM-MPI model. The memory usage can be monitored by some open source and public tools such as Valgrind etc.*

**Summary of changes:** We thank the reviewer for this suggestion. We extended section 5.2 with the memory footprint analysis.

*- In the validation section (5.1), author's point that the SHYFEM-MPI model is not bit-to-bit reproducible due to the order of the floating point operations. Although, it is not clear what kind of operations are mentioned in here (it would be nice to extend this part little bit), intel compiler provides way of ensuring order of the floating point operations such as in reduction operations via using special -fp-model precise flag. The special compile flag could slow the model around 10% but it could also help to achieve bit-to-bit results. Is this flag used when building model as well as PETSc library? Personally, two different execution of same configuration with same domain decomposition and same number of cores must produce identical results but it seems that SHYFEM-MPI model does not provide this capability, which is the main drawback of the current implementation. It also makes hard to couple the SHYFEM-MPI model with the other earth system model components (i.e. atmosphere model) because the non-linear interactions among the model components could lead totally different answer, which is very dangerous. The authors also discuss about a version that could create the reproducible results but I am not sure why it is just for debugging. I think that version needs to be publicly available. I think providing a model code to the ocean user community with a lack of reproducibility is very dangerous. I strongly suggest that the authors need to publish their reproducible version of the code in the code availability section. It would have some performance problems but at least it can be used safely in the scientific applications.*

**Summary of changes:** We thank the reviewer for pointing out this aspect that was one of the most discussed during our work. The intel compiler flag -fp-model is actually used when compiling the model but it is not enough to guarantee the bit-to-bit reproducibility when facing a parallel application. The -fp-model flag preserves the standard

order during the evaluation of a single floating point expression, but in a parallel application the order of the floating point operations depends also on the order in which the data are received from the neighbour processes. The not-deterministic order in the message exchange is mainly introduced by the not-blocking send/receive, by the use of ANYSOURCE rank specification during a receiving call and by certain implementation of the reduction operations. We obtained a bit-to-bit reproducible version of the model by removing all the sources of not-determinism; the counterpart is that in that version the performance drops dramatically. The parallel-SHYFEM software package that we published already supports the compilation of the model to be bit-to-bit reproducible with respect to the sequential run; the user can obtain the bit-to-bit version using the DEBUGON compiler key.

*Typos:*
*- Page 4 / Line 96: Referencing to the tables and figures must be consistent across the document. In the same cases the table and figure numbers are given inside the parenthesis and others not. Please review them carefully. For example, table (1) needs to be Table 1.*
*- Page 11 / Line 225: Please cite PETSc correctly using information in the following link, https://petsc.org/release/#citing-petsc*
*- Page 17 / Figure 6: I think there is something wrong with Figure 6. There is no top, bottom-left etc. So, the figure caption and figure needs to be consistent.*

**Summary of changes:** We thanks the reviewer for these corrections, all the typos were fixed in the revised version of the manuscript

---

## Author Response (AR2)

**SUMMARY OF CHANGES**

GIORGIO MICALETTO, IVANO BARLETTA, SILVIA MOCAVERO, IVAN FEDERICO, ITALO
EPICOCO, GIORGIA VERRI, GIOVANNI COPPINI, PASQUALE SCHIANO, GIOVANNI ALOISIO,
AND NADIA PINARDI

To the Topical Editors of Geoscientific Model Development

Dear Topical Editor,

The revised version of the manuscript GMD-2021-319 entitled "Parallel Implementation of the SHYFEM Model" has been edited to meet the suggestions included in the revision.

**1. REVIEW #1**

We thank the reviewer for good appreciation of the manuscript.

**2. REVIEW #2**

*I would like to thank to the authors to address most of my concerns about the previous version of the manuscript. Here are my comments about the current version of the manuscript.*
*What is the portion of the redistribution timing between PETSc and model when all execution time is considered? It might be nice to add this information to the text in Sec. 3.4.*

**Summary of changes:** The time needed for the redistribution between the PETSc and the model depends by the number of cores used for the simulation and its impact increases as the number of cores used increases ranging between 1% and 10% of the overall execution time.

*I am still not convinced about the parallel implementation of the model and its reproducibility issue. It does not makes sense to me to have a numerical model with reproducibility issue. This indicates a major problem in the parallelization approach used in here. Please discuss the reproducibility issue in a more detailed way (maybe a dedicated section could be good idea) by comparing with the other models? How this issue is addressed by other unstructured ocean models (i.e. SCHISM and others) used in the literature? If other models have also same issue, please add the relevant studies and papers as reference. If not, is this specific to optimized version of PETSc and/or used solver? If issue is related with the PETSc specifically, please add reference of other applications that indicates the similar issue. It seems that DEBUGON version of the model is able to reproduce the results with very poor performance. This indicates that there is no way to run model in*

*parallel and to create reproducible results. In same cases, poor performance can be acceptable if the configuration can not be run in single node due to the memory limitations but this is just for very high-resolution cases that requires lots of memory.*

**Summary of changes:** Section 5.1 was meant to include the discussion concerning the model validation and reproducibility, hence, instead of creating a new Section, we have further extended Section 5.1 to better point out the reproducibility issue and its origin. We better clarified that the round-off error comes from the floating point numbers representation and it is one of the main reason why all the numerical models, including GCMs, are affected by uncertainty of the results. We also extended the Section 5.1 citing other works regarding the assessment of models reproducibility when changing the computational environment such as compilers, HPC architectures or MPI decompositions. Namely, Cousins and Xue (2001) developed the parallel version of Princeton Ocean Model (POM) and found that there is a significant difference between the serial and parallel version of the POM concluding that the error from the data communication process via MPI is the main reason for the difference. Wang et al. (2007) studied the results of the atmospheric model SAMIL simulated with different CPUs and pointed out that the difference is chiefly caused by the round-off error. Song et al. (2012) assessed the round-off error impact due to MPI on the parallel implementation of Community Climate System Model (CCSM3). Guarino et al. (2020) presented the evaluation of the reproducibility of HadGEM3-GC3.1 model on different HPC platforms. Geyer et al. (2021) assessed the limit of reproducibility of COSMO-CLM5.0 model comparing the same code executed on different computational architectures.

Their analysis showed that the simulation results are dependent on the computational environments and the magnitude of uncertainty due to the parallel computational error could be in the same order as that of natural variations in the climate system.

Finally we underlined that although we can force the MPI version to execute the floating point operations in the same order of the sequential version and then the parallel model results are bit-to-bit identical with those of the serial model, we cannot guarantee the results of the serial model have no uncertainty, because the serial model also contains the round-off error. For this reason the "reproducible" version of MPI should be considered only for debugging purpose and it should not be meant as a more reliable version of the code w.r.t. the efficient MPI version.

*It would be nice to add information and figures that compare the optimized version of the code with the debug version. Please also include speedup (based on sequential code) information to the benchmark results. How long does it take the sequential run of same model configuration? This will give more information to the potential users about the overall performance of the parallel version of the model.*

**Summary of changes:** We enriched Fig. 11 reporting the performance comparison among the efficient version of the model, the execution without I/O and the debugging

version. We also added the detailed execution time in Table 3.

*The benchmark results indicates also issue related with the parallel I/O implementation that could be addressed in the future version of the model. Especially for high-resolution cases with high-frequency output, this could be a bottleneck and might affect overall performance of the modeling system. Using third-party libraries like PIO, SCORPIO or ADIOS might be a better approach rather than using in-house parallel I/O implementation since they are well tested and proved their scalability for different applications.*

**Summary of changes:** We thank the reviewer for these suggestions. In Section 4.4 we added the following sentence:

```
we are evaluating the adoption of efficient external libraries to enhance the
I/O performance in the next version of the model code.  Among the suitable I/O
libraries, we mention here XIOS, PIO, SCORPIO or ADIOS
```

*Typos/Figures:*
*- The multiple citations needs to be formatted correctly. For example, page 2, line 25 will be in following form (Casulli and Walters, 2000; Chen et al., 2003) and page 2, line 26 will be in (Danilov et al., 2004; Zhang et al., 2016; Umgiesser et al., 2004).*
*- Please combine Fig. 11 and 12 and present like that (red vs. blue etc.). It is hard to compare the benchmarking results (with and w/o I/O).*

**Summary of changes:** We thank the reviewer for pointing this out: we modified the citations format in order to be compliant with the reviewer suggestion. We also merged Fig. 11 and 12